# Emerging Roles of Microglia Depletion in the Treatment of Spinal Cord Injury

**DOI:** 10.3390/cells11121871

**Published:** 2022-06-09

**Authors:** Junhao Deng, Fanqi Meng, Kexue Zhang, Jianpeng Gao, Zhongyang Liu, Ming Li, Xiao Liu, Jiantao Li, Yu Wang, Licheng Zhang, Peifu Tang

**Affiliations:** 1Senior Department of Orthopedics, The Fourth Medical Center of Chinese PLA General Hospital, Beijing 100037, China; deng_junh@whu.edu.cn; 2National Clinical Research Center for Orthopedics, Sports Medicine and Rehabilitation, Beijing 100853, China; bonegaojp@163.com (J.G.); lzy_westlife@163.com (Z.L.); liming891215@163.com (M.L.); liuxiao23_314@163.com (X.L.); lijiantao618@163.com (J.L.); 3Department of Spine Surgery, Peking University People’s Hospital, Beijing 100044, China; fanqimeng@pku.edu.cn; 4Department of Pediatric Surgery, The Chinese PLA General Hospital, Beijing 100853, China; zhangkexue301@163.com; 5Beijing Key Lab of Regenerative Medicine in Orthopaedics, Key Laboratory of Musculoskeletal Trauma and War Injuries PLA, Institute of Orthopaedics, The Chinese PLA General Hospital, Beijing 100853, China; wangwangdian628@126.com

**Keywords:** spinal cord injury, microglia, cell typing, cell depletion

## Abstract

Microglia, as the resident immune cells and first responder to neurological insults, play an extremely important role in the pathophysiological process of spinal cord injury. On the one hand, microglia respond rapidly and gather around the lesion in the early stage of injury to exert a protective role, but with the continuous stimulation of the injury, the excessive activated microglia secrete a large number of harmful substances, aggravate the injury of spinal cord tissue, and affect functional recovery. The effects of microglia depletion on the repair of spinal cord injury remain unclear, and there is no uniformly accepted paradigm for the removal methods and timing of microglia depletion, but different microglia depletion strategies greatly affect the outcomes after spinal cord injury. Therefore, this review summarizes the physiological and pathological roles of microglia, especially the effects of microglia depletion on spinal cord injury—sustained microglial depletion would aggravate injury and impair functional recovery, while the short-term depletion of microglial population in diseased conditions seems to improve tissue repair and promote functional improvement after spinal cord injury. Furthermore, we discuss the advantages and disadvantages of major strategies and timing of microglia depletion to provide potential strategy for the treatment of spinal cord injury.

## 1. Introduction

Spinal cord injury (SCI) is a severe traumatic condition of the central nervous system (CNS) that can cause motor, sensory, and autonomic dysfunction below the level of injury [1,2]. Currently, the estimated annual global incidence of SCI is 40 to 80 cases per million population with an approximate three million SCI patients worldwide [3]. After SCI, more than 45% of patients will have permanent dysfunction, and spontaneous functional recovery is extremely limited. Moreover, due to a series of complex pathophysiological processes, the current effective treatment for SCI is still unavailable [4,5].

The pathophysiological process of SCI includes the primary SCI and secondary SCI. The primary SCI is mainly caused by the initial traumatic impact, resulting in the direct damage to the spinal cord tissue. It usually occurs within minutes to hours following the injury, characterized by the cell necrosis, hemorrhage, edema, etc. On the basis of the primary injury, SCI further develops within a few days to several weeks, causing a series of secondary injuries including inflammatory response, apoptosis, lipid peroxidation, and the production of a large number of free radicals. The secondary injury leads to massive neuronal and glial cell death and glial scarring, and eventually forms necrotic cysts that hinder nerve regeneration. Among them, the inflammation is considered to play an extremely important role in the SCI [6,7].

Inflammation after SCI is complicated and regulated by various types of cells, including microglia, astrocytes, neutrophils, monocytes, macrophages, and lymphocytes [8,9,10]. In the initial stage of SCI, various types of major inflammatory cells aggregate to the lesion, clear the damaged tissue by actively phagocytosing myelin and dead cell debris, and gather around the lesion core to limit the further expansion of the injury. However, with the progression of the SCI, the aggregation of the above-mentioned cells induces a strong immune response and secretes a large number of cytotoxic substances, such as free radicals, nitric oxide, and various inflammatory factors, including interleukin and tumor necrosis factor, etc., which further aggravates the injury of spinal cord tissue. As resident immune cells of the CNS, microglia play a major and critical role in the inflammatory response following SCI. Despite the current mainstream view that microglia are the main detrimental factor in the inflammatory response after SCI, there is still controversy as to whether microglia need to be removed after SCI for tissue repair and functional recovery. To better evaluate the role of microglia in the repair of SCI, especially after their removal, this review systematically summarizes the research progress of microglia depletion on SCI repair, with focuses on the effect of microglia depletion on SCI repair, the methods of microglia depletion, and microglia removal timing.

## 2. Overview of Microglia

### 2.1. The Origin and Physiological Role of Microglia

Microglia are resident immune cells in the CNS, accounting for approximately 5–15% of the total number of cells in the CNS [11,12]. They originate from myeloid progenitor cells in the yolk sac during the embryonic period and are a special type of mononuclear macrophage [13,14,15]. Under physiological conditions, microglia are in a “resting state” and characterized by small cell diameters and various morphologies, similar to neuronal structures with many tiny protrusions extending hierarchically from the cell body. While the cell body does not move much, the branches of microglia are highly dynamic, constantly stretching, moving, and therefore surveying the CNS every now and then [16].

As the first line of defense, microglia in the “resting state” were reported to directly or indirectly contact neurons, astrocytes, oligodendrocytes, etc. through their multi-level branches, and then play a dynamic surveillance role in the CNS [17,18]. Specifically, they are able to make a quick response to the potential pathological damage, then quickly and effectively remove invading pathogens and cell debris through phagocytosis to maintain the stability of the neural network of the CNS [19,20,21]. On the other hand, microglia function as a special kind of secretory cells, which can secrete trophic factors in the “resting state” to regulate the activity of neurons and oligodendrocytes, thereby affecting important processes such as axon growth and myelination of neurons [22]. The absence or dysregulation of such cells may directly lead to abnormal immune regulation, inflammatory storm, and neuronal death in the CNS [23,24]. Notably, the microglia in the spinal cord and thalamus exclusively express the ST2 marker, which is the specific receptor for interleukin-33 and mediates the microglial engulfment of synapses in these two regions. This biological process is responsible for the important circuit formation in the spinal cord [25,26]. Therefore, microglia play an important regulatory role in the CNS under physiological conditions, and at the same time provide an important immune surveillance system to maintain the development and homeostasis of the CNS.

### 2.2. The Role of Microglia in SCI

Microglia, as resident immune cells of the central nervous system, play a major regulatory role in the inflammatory response after SCI, but its regulatory mechanism on SCI remains unclear [27,28]. Following SCI, microglia are rapidly activated by various pathological factors and lead to a range of alterations in morphology and function. For cell morphology, there is an “amoeba-like” change, which is manifested by the rapid enlargement of the cell body, and the shrinkage of the elongated branch structures (they become shorter and thicker). For cell function, the proliferation and migration activities of microglia are significantly increased, and the phagocytic activity and secretion of cytokines are significantly enhanced. Although moderately activated microglia have a certain positive effect after SCI, those continuously stimulated by the injury often exhibit abnormal activation and exert toxic effects on neuronal cells, which promotes inflammatory cell aggregation and infiltration and further activates the inflammatory response in turn, thereby mediating secondary injury in the spinal cord [29,30].

Previous studies [31,32] believed that microglia released a large amount of harmful substances after SCI, resulting in a large number of neural cells death, axonal degeneration, demyelination, etc. [4], and simultaneously provoked astrocytes through paracrine effects and interacted together to form glial scars, to hinder axonal regeneration. Liddelow et al. [33] found that activated microglia could secrete cytokines such as IL-1a, TNF, and C1q in the central nervous system, which induced neurotoxic reactive astrocytes and then caused the neuronal and oligodendrocytes deaths. If any of the IL-1a, TNF, and C1q-encoding genes were conditionally removed on microglia using gene editing technology, animals would have shown much better motor functional recovery after SCI [34,35,36]. Moreover, the persistent activated microglia were reported to induce chronic neuro-inflammation, leading to the further tissue injuries and cell deaths, as well as cognitive deficits [37].

As for the subsets of microglia, activated microglia were originally considered to be homogeneous cells. But recently, a growing number of studies have revealed that after activation, microglia mainly have two cell subtypes with different cell markers and functions, namely the classical activated type (M1 type) and selective activated type (M2 type) (Figure 1). Pro-inflammatory factors, such as lipopolysaccharide and γ-interferon, can induce the transformation of “resting” microglia into M1 cells [38]. When SCI happens, microglia quickly recognize harmful stimuli via a series of immune receptors such as toll-like receptors and nucleotide-binding oligomerization domain- like receptors. The activated microglia then shift their function and polarize into being pro-inflammatory due to exposure to interferon-r and cell debris [39,40]. M1-type microglia are associated with tissue damage and pro-inflammatory responses, and secrete pro-inflammatory cytokines such as tumor necrosis factor-a (TNF-a), interleukin 1b (IL-1b), chemokines, antigen-presenting molecules, etc., which recruit inflammatory cells to the injured lesions, increase the phagocytosis of necrotic cells, and improve the defense clearance of the CNS against invasive stimuli to some extent [41]. However, M1-type microglia are usually over-activated, and then aggravate the downstream inflammatory response storm, resulting in increased neuronal cells death and tissue injury. Therefore, M1-type microglia are also called pro-inflammatory microglia.

Unlike the M1-type microglia, M2-microglia are mainly involved in tissue repair, the regulation of inflammatory response, neuronal differentiation, etc. [42,43]. They can be activated by cytokines such as IL-4 and IL-10 to become M2-type microglia, which mainly express IL-4, IL-10, and growth factors such as insulin-like growth factor 1 and transforming growth factor TGF-β, etc. They also have an enhanced ability to remove myelin debris and other necrotic substances, and are able to promote repair and functional recovery after SCI. Therefore, M2 microglia are also known as anti-inflammatory microglia. Previous studies have shown that M2-type microglia can be further classified into three subtypes: M2a, M2b, and M2c, which are stimulated by different factors and have relatively different functional phenotypes [44]. M2a cells are the main subtype of M2 microglia induced by IL-4 or IL-13, and express those neuroprotective markers, including CD206, Ym1, and ARG. -They mainly secrete anti-inflammatory factors, neurotrophic factors and etc. to promote tissue repair and reconstruction after SCI; M2b type, which usually appears under the stimulation of virus infection and immune complexes, is considered to be a transitional microglia between M1 and M2a, and shows both pro-inflammatory and anti-inflammatory effects at the same time; the M2c type, also a phenotype of neuroprotective microglia, is often stimulated and produced by IL-10 and TGF-b. This type of microglia usually shows high expression of TGF-b, SOCS3, and IL4R-a markers after SCI, which are mainly associated with synaptic remodeling, phagocytosis, and wound healing [45]. M2a and M2c are regarded as the major neuroprotective cell types of microglia, which can effectively limit secondary inflammation-mediated tissue damage and promote spinal cord repair. Moreover, the expression of the aforementioned neuroprotective markers of M2a and M2c were positively correlated with neurological outcome after SCI, especially at the early-stage post-injury [45,46].

With the deepening of research on microglia, the simple classification of microglia into M1 and M2 type were gradually considered to be incorrect. Instead, the activation into M1/M2 microglia was regarded as a dynamic process. Within the first few hours after SCI, microglia are initially activated to M2-type cells temporarily and play an anti-inflammatory role. They are manifested as enhanced phagocytic activity, accelerated clearance of irritants and cellular debris, enhanced secretion of anti-inflammatory factors, and neurotrophic factors, which could inhibit the excessive immune-inflammatory response and promote the repair of damaged tissue. In a few hours to several days after SCI, due to the continuous stimulation of the spinal cord, microglia rapidly proliferate and gradually transform from M2 cells to M1 cells, exerting neurotoxic effects on neurons via the secretion of inflammatory factors, activation of the downstream immune response, and further aggravation of the secondary injury of the spinal cord. After entering the subacute and chronic phases of SCI, M1-type cells become the major cell type of microglia, and continuously exacerbate the SCI and hinder tissue repair [47,48]. Therefore, how to minimize activation of M1-type cells and maintain the activation of M2-type cells in microglia will be one of the important directions of the treatment for SCI.

## 3. The Effect of Microglia Depletion on SCI Repair

As mentioned above, although activated microglia after SCI play a certain role in neuroprotection and repair at the early stage, excessively activated microglia simultaneously release a large number of harmful substances, further aggravating SCI and impeding the damage repair and functional recovery. Thus, microglia are required to be regulated to minimize their deleterious effects. What if microglia depletion is performed during SCI? Is it possible to suppress inflammatory storms and promote nerve regeneration and functional recovery?

### 3.1. The Adverse Effects of Microglia Depletion on SCI

At present, most studies have found that the depletion of microglia before SCI significantly inhibits the repair and functional recovery in animals with SCI [49,50,51,52]. Li et al. [49] found that microglia could promote long-distance axon growth after spinal cord crush injury and removal of microglia will directly affect the process of nerve regeneration in neonatal mice with SCI. Furthermore, single-cell sequencing revealed that microglia can temporarily secrete fibronectin and its binding proteins to provide the extracellular-matrix bridge, thereby promoting spinal cord axon regeneration. Victor Bellver-Landete et al. [50] also found that microglia were an essential component of protective scars after spinal cord contusion injury, and the depletion of microglia directly led to decreased secretion of insulin-like growth factor-1, disordered scar structure, and ultimately hindered functional recovery in animals. Fan et al. [51] found that even M1-type microglia/macrophages were still able to promote spinal cord repair by inhibiting astrocyte activation after spinal cord contusion. Therefore, if the microglia had been completely removed, reactive astrocytes would then have been activated, which would further disrupt tissue repair and nerve regeneration. Consistently, Hakim et al. [52] showed that in mice with spinal cord contusion microglial activation exerted a strong positive effect on SCI repair as activated microglia were responsible for the removal of cell debris and immune-regulatory effects on inflammation and astrocytic response. Similarly, our previous study [53] also showed that 5-week-long microglia depletion would cause glial scar disorder, delay astrocyte aggregation, further axonal degeneration, and functional impairment on mice with spinal cord crush injury.

While the methods and strategies of microglia depletion in the aforementioned studies were different, all of them found that microglia depletion would further affect the repair after SCI. Thus, microglia depletion might not improve functional recovery after SCI, but even further aggravate injury on animals in these cases.

### 3.2. The Beneficial Effects of Microglia Depletion on SCI

Obviously, those above findings contradicted the scientists’ original assumption that the complete elimination of hyper-activated microglia should lead to beneficial outcomes. To this end, the researchers further explored whether microglia depletion could have beneficial effects? Victor Bellver-Landete et al. [50] changed their previous microglia depletion strategy by removing microglia three weeks before the injury and stopping microglia intervention after injury. The results showed that microglia depletion did not significantly affect SCI, but again no beneficial outcomes were found. Consistent with Victor’s study, Li et al. [54] found no locomotion functional recovery after spinal cord contusion via either pre- or post-injury microglial depletion for six weeks. However, their study demonstrated that microglial depletion significantly reduced the production of reactive oxygen species and improved neuronal survival and neurological recovery, including cognition and depressive-like behavior. Igor Jakovcevski et al. [55] pointed out that if microglia were continuously removed from 2 weeks before SCI to 2 weeks after spinal cord compression injury, the removal of microglia in the early stage of the injury (1 week after injury) could promote functional improvement after SCI, while in the late stage of the injury (5 weeks after injury), there was no significant improvement in functional outcomes after SCI even though there were some improvements in gliosis and cholinergic nerve regeneration at that time. Interestingly, Gaëtan Poulen et al. [56,57] reported that, regardless of rodents or non-human primates, a transient depletion of microglia could significantly enhance functional recovery with the alleviation of tissue damage, whereas prolonged microglia depletion was not beneficial for tissue repair after spinal cord hemisection. Ma et al. [58] found that two-week-long microglia depletion significantly improved the local microenvironment after SCI in rats, and enhanced the electrophysiological activity and functional recovery of animals. And if combined with their self-designed hydrogel-based scaffold, motor function recovery could be further improved after the complete transection spinal cord injury.

Taken together, the effects of microglia depletion on SCI are still controversial. Different microglia depleting approaches, microglia depleting periods, and SCI models directly determine the positive or negative outcomes after SCI. Notably, a growing body of evidence shows that the persistent microglia removal seemed to be not conducive to tissue repair after SCI, while selective and strategic removal of microglia might improve recovery after SCI.

## 4. The Strategy and Timing of Microglia Depletion

As previously mentioned, different microglia depletion strategies greatly affect the outcome of tissue repair after SCI. At present, there is no unified and recognized paradigm for microglia depletion. Each research team applied respective strategies of microglia depletion for special research purposes.

### 4.1. The Main Method of Microglia Depletion

The methods of microglia depletion can be mainly divided into two categories: gene manipulation and small-molecule intervention. The former mainly uses microglia-specific surface markers such as Cx3Cr1, TMEM119, Sall1, etc., to construct specific microglia transgenic animals and then achieve the conditional removal of microglia; the latter involves the design and construction of small-molecule compounds against specific microglia cell markers so as to induce apoptosis and necrosis of microglia and then deplete these cells.

#### 4.1.1. Microglia Depletion via Gene Manipulation

Microglia depletion via gene manipulation usually requires the construction of specific transgenic mice combined with specific toxic substances, such as diphtheria toxin (DT) or herpes virus, to conditionally remove microglia. Parkhurst et al. [59] constructed Cx3cr1^CreER^: Rosa26^iDTR^ transgenic mice and achieved conditional knockout of microglia through the CreERT system and DTR system. Specifically, they first constructed Cx3cr1^CreER^ mice that specifically express Cre recombinase and estrogen receptor (ER) at the site of Cx3cr1^+^ cells and obtained Cx3Cr1^CreER^: Rosa26^iDTR^ transgenic mice by crossing Cx3Cr1^CreER^ mice with Rosa26^iDTR^ mice. Then, they induced Cre recombinase expression by tamoxifen, and then injected DT intraperitoneally to achieve Cx3cr1^+^ cells knockout. Considering that peripheral monocyte-macrophages also express Cx3cr1 protein but with more frequent cell turnover (about 7 days), the intraperitoneal injection of DT would be performed 30 days after tamoxifen induction to achieve specific microglia depletion. Similarly, CX3CR1^CreER^: Rosa26^iDTA^ [60], Cx3cr1^Cre^:Csf1r^fl^ mice [49], and CD11b^HSVTK^ transgenic mice [55] have been constructed successively to conditionally remove microglia in the CNS. They either used the CreERT system, the Cre-Flox system, or expressed suicide genes to achieve the depletion of microglia.

The advantage of microglia depletion via gene manipulation includes: (1) it can manipulate the time series to achieve microglia depletion at a specific time point or a specific period, allowing the experimenter to flexibly manipulate the observation window; (2) In addition, microglia-specific knockout can be achieved using the self-renewal characteristics of microglia. The disadvantage of microglia depletion at the genetic level is that: (1) it mainly relies on transgenic mice. The construction and feeding cycle are time-consuming, and special identification is also required; (2) At present, the construction of microglia transgenic animals is mainly carried out on mice, since it is difficult to expand to rats or other mammals; (3) Microglia depletion via gene manipulation is not of very high efficiency (about 70%) and cannot be maintained for a long time, since the renewed microglia do not express Cre recombinase. The application of tamoxifen and diphtheria toxin may affect the survival of mice or exacerbate function of mice with SCI.

#### 4.1.2. Microglia Depletion via Small-Molecule Compounds

Unlike genetic manipulation to remove microglia, microglia depletion via small-molecule compounds is more convenient and diverse. They are either designed to impress those targets important for microglial survival, such as colony-stimulating factor 1 receptor (CSF1R), or they exert direct toxic effects leading to microglial cell death. For example, CSF1 is an extremely important regulator of microglia/macrophage proliferation, differentiation, and survival, so interventions on CSF1R can achieve targeted depletion of microglia. The small-molecule compound PLX3397 [14,33,61] and its second-generation product PLX5622 [50,62,63] are currently widely used specific inhibitors of CSF1R. They are highly specific to CSF1R and able to penetrate the blood–brain barrier and competitively occupy the site of CSF1, resulting in microglial cell death with up to 99% effectiveness. At the same time, studies have shown that PLX small-molecule compounds had little effect on other types of cells and animal behavior in mice, and thus are considered very good microglia-targeted regulation drugs.

In addition to these two star products, there are other CSF1R-specific inhibitors (GW2580 [64], BLZ945 [65], etc.), chlorophosphate liposomes [66], and M1-type microglia targeted depletion drugs such as gadolinium chloride [67], which could achieve microglia depletion to a varying degree. GW2580 is an orally selective inhibitor of the tyrosine kinase activity of CSF1R, which could selectively inhibit microglia/monocyte proliferation [64]. BLZ945, another brain-penetrant CSF1R inhibitor, could decrease the number of microglia by approx. 60% after its application [65]. Chlorophosphate liposomes can inhibit ADP/ATP transport in microglia and induce cell death by generating non-hydrolyzable ATP analogs, while gadolinium chloride competitively inhibits the active calcium activity of M1-type microglia and destroys the endoplasmic reticulum membrane to induce the death of microglia.

Different small-molecule compounds have distinct characteristics and modes of action, and the efficiency of microglial depletion is also different. Therefore, they should be selected according to their specific experimental purposes. The advantages of small-molecule compounds are very obvious: (1) It is not dependent on the species of animal. Most animals depend on CSF1 for microglial survival and proliferation. Taking PLX products as an example, PLX small-molecule compounds can theoretically work on all animals; (2) Small-molecule compounds can be taken orally with high selectivity, and some of them can pass through the blood–brain barrier, which is very convenient for application; (3) They are relatively safe, as they do not affect the stability of animal genomes. However, the shortcomings of small-molecule compounds are also obvious. For example: (1) It has been reported that [68] CSF1R inhibitors are not microglia-specific. Instead, they affect both hematopoietic and macrophage functions; (2) Manipulation at specific time points is unavailable to small-molecule compounds, and the starting point of the specific action on microglia is difficult to determine.

Taken together, when it comes to microglia depletion, there are two main strategies: gene manipulation and small-molecule intervention. Both strategies have their advantages and disadvantages and could be combined to make up for their respective shortcomings and achieve specific purposes (Table 1).

### 4.2. The Timing of Microglia Depletion

The optimal timing for microglia depletion remains controversial at present. The different purposes or approaches to microglia depletion also influence the timing selection of microglia depletion (Table 2).

Taking microglia depletion via gene manipulation as an example, some studies performed diphtheria toxin application 7 days after tamoxifen induction [72] and found that microglia were specifically reduced and lasted for a week. However, in theory, the peripheral monocytes and macrophages would also be specifically removed at this time. On the other hand, in order to obtain selective microglia depletion, many studies chose to use diphtheria toxin 3–4 weeks after tamoxifen induction [59,63], since the peripheral macrophages would have completed the cell turnover and no longer express the diphtheria toxin receptor, whereas microglia still expressed DT receptor. However, the time frame for microglia depletion via gene manipulation is relatively short and studies have shown [29] that the number of microglia return to normal one week after the application of diphtheria toxin.

As for the timing of microglia depletion via small-molecule compounds, it is more flexible and diverse. Taking the most widely used PLX3397 and PLX5622 as examples, most studies chose to apply CSF1R inhibitor three weeks or more before the injury [50,60] and maintain the drug to the end of the experiment. Additionally, some chose two weeks or one before the injury to remove microglia [33,49,53,73]. Furthermore, some studies on microglia depletion started on the day of SCI [58], or 3 days after SCI [50]. However, regardless of the timing, the results of the study showed that the number of microglia decreased significantly after the application of PLX CSF1R inhibitors and the number of microglia returned to normal levels after drug withdrawal. Taking into consideration the practical clinical application, it seems to be more reasonable if microglia depletion is obtained after injury, but the specific application period still needs further research to determine.

In conclusion, there is no unified timing of microglia depletion during SCI, and different strategies or purposes of microglia depletion determine the actual intervention timing. In terms of the length of the intervention time compared to the depletion of microglia for the entire experimental time window, the short-term removal of microglia seems more reasonable as microglia indeed play positive roles in the CNS [56].

## 5. Conclusions

Microglia play a very important role in normal physiological processes and after SCI. Following SCI, on the one hand, microglia respond quickly and play a certain role in neuroprotection and repair by removing invading pathogens and cell debris through phagocytosis. On the other hand, continuous activated microglia often exert adverse effects on neuronal cells, promote inflammatory cell aggregation and infiltration, and further activate the inflammatory response in turn, thereby mediating secondary injury in the spinal cord.

The effects of microglia depletion on SCI are still controversial, and different microglia depletion strategies greatly affect the outcome of tissue repair after SCI. Current studies have found that continuous microglia depletion is not conducive to tissue repair after SCI, and selective short-term depletion of microglia could effectively improve tissue repair and promote functional recovery after injury. Regarding the removal method and removal timing of microglia, there is still no uniform paradigm. Different strategies or purposes of microglia depletion determine the actual intervention timing and ultimately affect the outcome of animal SCI. Future research should focus more on: (1) How to achieve longer lasting microglia depletion via gene manipulation; (2) Exploring new and more microglia-specific small molecule compounds; (3) Identifying the optimal timing and duration of action of microglia depletion and elucidate its underlying mechanism. It is hoped that with the development of various new technologies and in-depth research on microglia, people can finally achieve the dynamic and selective regulation of microglia and make a breakthrough in the treatment of SCI in the future.

## Figures and Tables

**Figure 1 cells-11-01871-f001:**
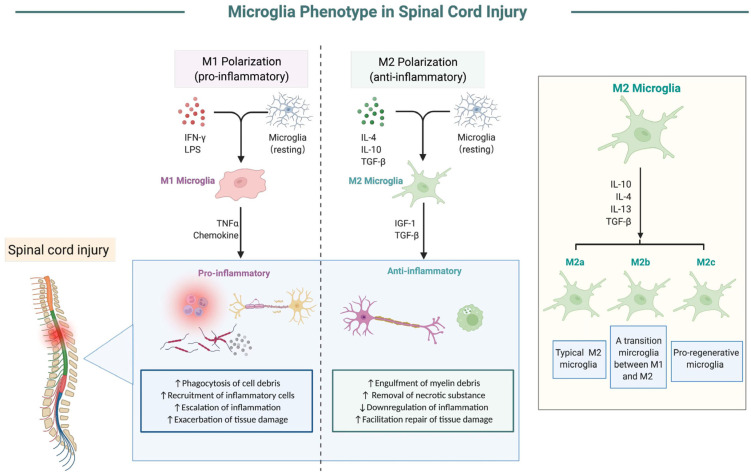
The polarization process of microglia after SCI.

**Table 1 cells-11-01871-t001:** The different strategy of microglia depletion and their respective characteristics.

Category	Strategy	Advantages	Disadvantages	Reference
Gene manipulation	CreERT system	Precise timing control; Microglia-specific removal;	Need for tamoxifen and diphtheria toxin induction; Relatively time-consuming; Species limitations;	Hakim et al. [52]; Parkhurst et al. [59].
	Cre-Flox system	Easy to operate, and microglia are removed after birth; No need for tamoxifen and diphtheria toxin induction;	Difficult to achieve precise timing control; Possible adverse effects on peripheral mononuclear macrophages; Relatively time-consuming; Species limitations;	Li et al. [49]; Qin et al. [69].
	HSVTK system	No need for tamoxifen or diphtheria toxin induction;	Difficult to achieve precise timing control; Possible adverse effects on peripheral mononuclear macrophages; Relatively time-consuming; Species limitations;	Igor et al. [55]; Heppner et al. [70]; Varvel et al. [71].
Small-molecule compounds	CSF1R inhibitors	Oral or injectable administration and some can pass through the blood–brain barrier; Relatively time-saving; No species limitation;	Not specific to microglia and affect hematopoietic and macrophage functions; Difficult to achieve precise timing control;	Elmore et al. [14]; Lei et al. [68].
	Chlorophosphate liposomes	Local microglia depletion; Relatively time-saving; No species limitation;	Need to inject into a specific site in the CNS so increase the risk of secondary damage; Not specific for microglia.	Asai et al. [66].
	Gadolinium chloride	Selectively remove M1-type microglia without affecting M2-type microglia; Relatively time-saving; No species limitation;	Need to inject into a specific site in the CNS so increase the risk of secondary damage.	Miron et al. [67].

**Table 2 cells-11-01871-t002:** The outcomes and therapeutic effects of the different timing of microglia depletion on SCI.

Depleting Approaches	Depleting Timing	Model	Outcomes	Therapeutic Effects	References
CreERT system	Microglia were depleted after Tamoxifen injection	Contusion	Harmful	Affect function recovery following injury	Hakim et al. [52].
Cre-Flox system	Microglia were depleted after birth	Crush	Harmful	Impair bridge formation and axon regeneration	Li et al. [49].
HSVTK system	2 weeks before SCI and 2 weeks post SCI	Compression	Beneficial	Improve locomotor recovery in the early phase post SCI, but does not affect recovery in the following 4 weeks	Igor et al. [55].
PLX5622	3 weeks before SCI to 5 weeks post SCI	Contusion	Harmful	Worse functional recovery	Victor et al. [50].
PLX5622	1 day post SCI to 6 weeks post SCI	Contusion	Beneficial	Enhance cognitive behavior but failed to improve locomotion function	Li et al. [54].
PLX3397	Pregnant mice received PLX from E14 and newborn pups continued to receive PLX 2 weeks post SCI	Crush	Harmful	Impair bridge formation and axon regeneration	Li et al. [49].
PLX3397	1 week before SCI to 4 weeks post SCI	Crush	Harmful	Impair locomotor function and exacerbate tissue damage	Fu et al. [53].
PLX3397	0 day post SCI to 2 weeks post SCI	Transection	Beneficial	Enhance the electrophysiological activity and functional recovery	Ma et al. [58].
GW2580	4 weeks before SCI to 6 weeks post injury	Hemisection	Beneficial	Improve the fine motor function and decrease the gliosis	Gerber et al. [64].
GW2580	0 day post injury for 1 week (mice) or 2 weeks (non-human primates)	Hemisection	Beneficial	Promote functional recovery and tissue repair	Poulen et al. [56].
GW2580	0 day post injury for 6 weeks	Hemisection	Not beneficial	No impact on functional recovery	Poulen et al. [58].
GdCl_3_	1 day post SCI for once	Contusion	Harmful	Activate astrocytes and disrupt tissue repair and nerve regeneration	Fan et al. [51].

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
