# Peer review of "Emerging Roles of Microglia Depletion in the Treatment of Spinal Cord Injury"

_cells, 2022, doi:10.3390/cells11121871_

Round 1

Reviewer 1 Report

The present review manuscript “Emerging roles of microglia depletion in the treatment of spinal cord injury” by Junhao Deng from the Laboratory of Peifu Tanga at the Medical School of Chinese in Beijing is related to an important question in the field of spinal cord injury (SCI) i.e. the manipulation of the microglial response in the context of a spinal cord lesion. There is indeed a strong need to investigate the effect of therapeutic strategies that may promote functional recovery after SCI.

The aim of this review was to investigate outcomes of microglia depletion in SCI. Authors first started with a general introduction about microglia in SCI followed by the effect of their depletion (beneficial/detrimental), the strategy (gene manipulation/pharmacological) and finally the timing. The overall review is informative, unfortunately, recent literature on microglia depletion is missing to accurately present an overview of microglia depletion in SCI, in particular regarding beneficial outcomes of the depletion.

It concerns both genetic models and pharmacological approaches (including nonhuman primates), as well as the timing of depletion and transcriptional changes induced by microglia depletion (see DOI below).

These studies need to be presented and discussed. Also, partial versus complete microglia depletion may be interesting to discuss.

P181 The sentence is not correct if it not stated that it is a complete depletion of microglia.

Lines 174-178: the sentence needs revision.

A figure focusing on the timing of depletion, at least for the pharmacological approach would be very informative.

References must be included in table1.

Why to thank a facility in the fram of a review paper? “the Orthopedics Institute of Chinese PLA for providing the necessary experimental facilities in this study”.

"

doi.org/10.1523/JNEUROSCI.0860-21.2021: “Spinal Cord Injury Induces Permanent Reprogramming of Microglia into a Disease-Associated State Which Contributes to Functional Recovery”

doi:10.7150/thno.49199 “Delayed microglial depletion after spinal cord injury reduces chronic inflammation and neurodegeneration in the brain and improves neurological recovery in male mice”

DOI: 10.7150/thno.61833  «Inhibiting microglia proliferation after spinal cord injury improves recovery in mice and nonhuman primates »

DOI: 10.3390/brainsci11121643; “ Unlike Brief Inhibition of Microglia Proliferation after Spinal Cord Injury, Long-Term Treatment Does Not Improve Motor Recovery “

Reviewer 2 Report

This paper thoughtfully reviews  current information regarding the role of microglia depletion in the treatment of SCI. The manuscript is strengthened by complex description of methods used for microglia depletion via gene manipulation and small-molecule compounds, and discuss the timing of microglia depletion. However, the manuscript in the current version shows some weakness in in-depth data analysis in section 1.2 The role of microglia in spinal cord injury. I also miss current literary regarding in vivo SCI studies. Moreover, inflammation response depends on the type of SCI. Please specify which spinal cord model was used when quoting a specific findings from an in vivo studies. The authors are suggested to address all comments and suggestions provided by the review process.

Lines 35-36

Please update information regarding the SCI patients worldwide and new cases reported per year.

Lines  (33, 70)

Although the abbreviations for spinal cord injury and central nervous system (SCI and CNS) are explained in the first two chapters (lines 33 and 70), the authors mostly use their full text (lines 40, 81, 84, 89, 90, 92, 94, ...) in the MS.  Please correct.

Lines (80, 120, 123, 130)

It is more convenient to specify  the types of cells and add the citation than to use “etc.” (line 80). Similarly, please specify another  mechanisms known for M2c microglia  (lines 120) and  expressed molecules (lines 123 and 130) and add the citations.

Line 91

Please correct ...system f to maintain...

Lines 105-118

Although this paragraph is clear  and interesting, it deals with the activation of M1 microglia only under some in vitro conditions. Please  update this information and add data showing M1 activation in spinal cord after SCI (compression, contusion,...) in vivo. I also miss citations  after the sentence ...Pro-inflammatory factors such as (lipopolysaccharide and g-interferon...

Lines 126-135

In this part I miss new data regarding  the role of neuroprotective microglia (M2a- M2c phenotypes) in post-SCI repair. In addition, the sentence” Recent studies have shown that M2-type microglia... ” should be corrected, because results of David and Kroner, 2011 are not recent.

Lines 105-152

In general, this section can benefit from including a logic flow of microglia activation after SCI.

Line 174-177

Reconsider the two sentences. Please correct ...promote dspinal...  and also ...had been completed removd, reactive Astrocytes...

Lines 200-206

The authors comment their unpublished data.

Please briefly comment that activation of neuroprotective microglia is crucial for functional outcome after SCI. These recently published data would give a more balanced view.

Line 330

Please correct …promot…

Reviewer 3 Report

The submitted review is relevant and corresponds to the aim and scope of the journal “Cells”. Comments. 1.    Section "1.1. The origin and physiological role of microglia" should be supplemented with recent references (2019-2022) and moreover, with publications related not only to the brain, but also to the spinal cord. 2.    Please, correct errors on p.4, s.176 ("...the microglia had been completed removd,..."). 3.    The results of the minocycline use to inactivate microglia seem similar to transient microglia depletion, and, thus, should be added to the section "3.1.2. The microglia depletion via small-molecule compounds".

Reviewer 4 Report

Here I present my comments on the work entitled “Emerging roles of microglia depletion in the treatment of spinal cord injury“ presented by Deng et al. For publication in Cells (Manuscript ID: cells-1713503)

This review is well written and presented in a comprehensive manner. Also, the figure and the table are displayed clearly and in a comprehensive manner. From what I see, the topic and propositions given by the authors are sound and respond to a critical issue of Spinal Cord Injury (SCI). In principle, I feel enthusiastic when I see that the role of microglial cells on SCI is highlighted from a translational perspective.

Nervelessness, I would kindly require to Author‘s manuscript to include some new information on critical issues which I see as important to publish the review.

  1. The argumentation on M1- and M2-type microglia is true, but I feel that the cited literature is poor in order to support the Author‘s statements. For instance, in page 3 the number of specific citations is very small. Please include new literature supporting each of the key statements.

  1. Also table 1 must include the literature supporting different categorizations. Please add a new column where specific literature is cited for each of the advantages/disadvantages. I can see that some literature is cited in point 3.1.2. but clearly, there is a large room for connecting better with state-of-the-art literature on the topic.  

Round 2

Reviewer 1 Report

Authors have addressed all my questions.

With the use of GW2580, conversely to PLX, it is not a general microglia depletion but the inhibition of microglia proliferation, it should be mentioned in the text.

Minor : on page 6 the reference with long term inhibition of microglia proliferation is not well inserted. It appears as ref 76 at the end of the reference list.

Author Response

Dear reviewer,

Thanks very much for your helpful comments on the second version of manuscript. We do appreciate for your kind help and patience. 

As for the use of GW2580, we agree that we should mention it as an microglia inhibitor in the manuscript. Therefore, in the Page 7, section 3.1.2, we added the following sentences. "...which could achieve the microglia depletion to a varying degree. For the GW2580, it is an orally selectively inhibitor of the tyrosine kinase activity of CSF1R, which could selectively inhibits microglia/monocytes proliferation [64]. For the BLZ945, also as another brain-penetrant CSF1R inhibitor, could decrease about 60% number of microglia after its application [65]."

As for the reference, we have re-checked the reference [57] and reference [76], and totally agree with your opinion. We now have added the reference 76 to the right place, and corrected reference [57] and [76] into form “[56,57]”. The revised sentences were listed below.

"Interestingly, Gaëtan Poulen et al. [56,57] reported that regardless of rodents or non-human primates, a transient depletion of microglia could significantly enhanced functional recovery with the alleviation of tissue damage, whereas prolonged microglia depletion was not beneficial for tissue repair after spinal cord hemisection. "

And we also double-checked the manuscript to avoid another references-inserting mistakes. Thanks again for your kind help and patience.

Best regards,

Peifu.

Reviewer 2 Report

The MS by Junhao Deng et al, entitled „Emerging roles of microglia depletion in the treatment of spinal cord injury“ and submitted to Cells has been sufficiently improved to warrant publication in Cells.

I have only one comment:

Page 4

The authors should correct „lime“ to „limit“ in the following sentence: M2a and M2c were regarded as the major neuroprotective cell types of microglia, which could effectively lime secondary inflammation-mediated tissue damages and promote spinal cord repair.

Author Response

Dear reviewer,

Thanks very much for your helpful comments on the second version of manuscript. We do appreciate for your kind help and patience. 

As for the Page 4, we thanks for pointing this out, and we now have corrected it into the right spelling. And we also double-checked the manuscript to avoid another spelling mistakes. Thanks again for your help.

Best regards,

Peifu

Reviewer 4 Report

The authors have addressed satisfactorily all my concerts. 

Author Response

Dear reviewer,

Thanks very much for your helpful comments on the second version of manuscript. We do appreciate for your kind help and patience. 

Best regards,

Peifu